# Microwave Curing Characteristics of CFRP Composite Depending on Thickness Variation Using FBG Temperature Sensors

**DOI:** 10.3390/ma13071720

**Published:** 2020-04-07

**Authors:** Heonyoung Kim, Donghoon Kang, Moosun Kim, Min Hye Jung

**Affiliations:** 1Railroad Safety Research Team, Korea Railroad Research Institute, Uiwang 16105, Korea; hykim@krri.re.kr; 2School of Mechanical Engineering, Yonsei University, Seoul 03722, Korea; 3Urban Transit Research Team, Korea Railroad Research Institute, Uiwang 16105, Korea; mskim@krri.re.kr; 4International Carbon R & D Center, Korea Institute of Carbon Convergence Technology, Jeonju 54853, Korea; minhjung@kctech.re.kr; 5Department of Organic Materials & Fiber Engineering, Jeonbuk National University, Jeonju 54896, Korea

**Keywords:** microwave curing, cure monitoring, fiber Bragg grating (FBG), carbon fiber reinforced plastics

## Abstract

Microwave curing technology, which has seen increased commercialization recently due to its ability to cut the curing time and ensure high quality, requires an understanding of the curing characteristics of composite materials of varying thickness. Therefore, this study aimed to perform cure monitoring to evaluate the effects of variations in thickness on the quality of microwave curing. For this study, a fiber Bragg grating sensor was used to measure temperature changes in specimens during the curing cycle for cure monitoring which is generally used for optimization of the curing cycle; then, the time taken for temperature increase and overshoot of the specimen, and the times at which the specimen thickness varied, were quantitatively evaluated. Testing confirmed that microwave curing reduced the curing time in the sections in which the temperature rose; also, the specimen thickness caused overshoot of up to approximately 40 °C at the side, which can affect the curing quality of the composite materials. Furthermore, voids were observed on the side of all specimens. The results indicated that, in order to improve the quality of microwave curing of composite materials, the curing cycle should be optimized by considering the characteristics of the microwave curing equipment.

## 1. Introduction

Due to their high specific strength and specific stiffness, carbon fiber reinforced composite materials have attracted much research attention as substitutes for metallic materials in diverse areas, such as the aerospace [1,2], machinery, automobile, and railway industries [3,4]. The use of such materials is quickly expanding to other areas, including household items such as carrier cases, which need to be lightweight. However, due to the high manufacturing cost and power consumption occasioned by their long curing time, conventional curing methods for high-quality composite materials, such as the autoclave method, have been applied only in limited areas such as the aerospace and defense industries. Although other curing methods can reduce the manufacturing cost and power consumption by shortening the curing time, they also result in lower quality, and thus are mostly applied to products for which high quality is not a prime consideration. As such, interest in microwave curing, which can cut the curing time while ensuring the high quality of the composite materials, has been gradually increasing lately. The microwave curing method cures composite materials by using microwaves to irradiate a specimen, directly heating the selected material. The principle of microwave curing is that charge displacement occurs under an alternating microwave field. Then, oscillation of the particles generates heat in the composites based on the dielectric materials [5].

This process, because it directly heats the specimen, can greatly reduce the curing time compared to autoclave curing, which indirectly heats a specimen through convection of heated air. Moreover, microwave curing can also reduce total power consumption for composite manufacturing, in which the size of the structure increases more and more in comparison to other manufacturing methods. Therefore, there are many ongoing efforts to develop dedicated composite curing equipment that uses microwaves, and to commercialize the process [6].

In previous studies on the microwave curing of composite materials, 2.45 GHz was the most popular frequency in a commercial device for the curing of composite materials; since then, this frequency band has been the most common one used in microwave curing [7,8,9]. Based on a study, Rao [10] cured bifunctional epoxy matrices using microwaves and confirmed the uniform temperature distribution in the specimen. Papargyris [11] reported that microwave curing could reduce the curing time by 50% compared with conventional curing methods, and Sung [12] analyzed the relationship between the dispersibility and heat generation of MWCNTs, proving that MWCNTs improved heat generation by microwaves. In addition, Li [13] confirmed that heat transfer differed according to the thickness of stacked layers when curing carbon fiber composite material with microwaves. Kwak [14] showed that a carbon fiber composite material manufactured with microwave curing showed increased compression strength of about 82% compared to conventional curing methods. Xu [15] confirmed that, compared to thermal curing, using microwaves to cure carbon fiber composite materials decreased the curing time by 39%, while increasing the compression strength by 22%. Moreover, that study compared the strengths of specimens manufactured by autoclave curing and microwave curing [16].

Concerning the power consumption of commercial devices such as autoclaves and microwave equipment, Figure 1 presents the advantages of microwave curing. In this, power consumption per effective working volume of all devices is represented by heating power. For the analysis of comparison, conditions such as the same curing cycle and constant powers are applied to exclude fluctuation of power during the composite curing [17]. Of these, the circulating fan motor power consumption of the autoclave is added. It can be seen in the figure that, compared to that of autoclave curing, power consumption per volume of microwave curing rapidly decreases as the total volume of the manufacturing increases. When changing devices from device 1(VHM 180/200) to device 2(VHM 180/300, Vötsch Industrietechnik corp., Reiskirchen, Germany) in the case of microwave curing, power consumption per volume decreases from 15.5 kW/m^3^ to 9.7 kW/m^3^, a reduction of 37.2%. However, the reduction amount for autoclave curing (device 1: CGF1530, device 2: CGF2560, Changzhou Sinomac Machinery Technology corp., Changzhou, China) is only 13.4%, even though the absolute volume of each device is different between the two curing methods. This is a key point of microwave curing: it is a green technology for composite manufacturing.

However, when microwave curing is applied to a thick laminate, it is necessary to consider the effect of composite laminate thickness on manufacturing quality, even though microwaves can directly penetrate through composite material for heating the materials. As the thickness of a composite material increases, irregularities like voids form inside the material due to the limited depth of the microwaves. This can in turn lead to local overheating and cause overshoot, affecting the curing quality. Therefore, an accurate understanding of the effects of varying the thickness of a composite material on the quality of microwave curing can be used to optimize the curing for specific requirements. Cure monitoring is generally used to measure temperature and residual strain of a specimen and determine the optimal curing cycle. Previous studies on optimizing autoclave curing progressed from studies on cure monitoring using an electric sensor, such as a thermocouple [18], to studies using a fiber optic sensor to minimize the effect of the sensor on the specimen [19,20,21]. However, the use of an electric sensor in microwave curing is limited due to the reaction of the metallic material to microwaves, and thus some studies have used an IR thermal imaging camera or a fiber optic sensor [22].

This paper analyzed the effects of laminate thickness on the manufacturing quality of composites through cure monitoring performed using microwave curing equipment. For this purpose, temperatures of the laminate in several layers were measured using embedded FBG temperature sensors and an optical microscope was used to look at surface morphology and analyze the influences of laminate thickness.

## 2. Demodulation Principle of an FBG Sensor

The signal pattern of an FBG sensor can be expressed as the reflection of the Bragg wavelength (*λ_B_*) in Bragg gratings, as shown in Equation (1), and as changes of Bragg wavelength according to the temperature change and strain, as shown in Equation (2):(1)$$\lambda_{B}=2n_{e}\Lambda$$
(2)$$∆\lambda_{B}=\lambda_{B}\left[\left(\alpha_{f}+\xi_{f}\right)\Delta T+\left(1-p_{e}\right)\varepsilon \right]$$
where ‘ε’ is the strain, ‘Δ*T*’ is the temperature change, ‘$\alpha_{f}$’ is the thermal expansion coefficient, ‘$\xi_{f}$’ is the thermo-optic coefficient, and ‘$p_{e}$
’ is the photo-elastic constant.

If there is no change of strain (ε = 0), Equation (2) can be converted to Equation (3), which can be used for the temperature sensor:(3)$$\Delta T=T-T_{initial}=\frac{1}{\alpha_{f}+\xi_{f}}\frac{∆\lambda_{B}}{\lambda_{B}}$$
where ‘$T_{initial}$’ is the initial temperature of the data form the reference temperature sensor.

To manufacture an FBG temperature sensor as described above, we isolated the Bragg grating using a capillary glass tube, as shown in Figure 2, so that there was no effect of strain. The diameter of the capillary tube was 128 μm, i.e., slightly larger than 125 μm, which was the outer diameter of the optical fiber cladding [23].

Packaging the FBG sensor in various forms based on the basic principle described above makes it possible to measure physical quantities such as displacement, acceleration, load, etc. Many such packaged commercial products are currently available on the market.

## 3. Cure Monitoring Test of Composite Material Using FBG Sensors

To check the curing characteristics of the composite material according to the specimen thickness, three parameters were specified for each case. Three types of laminates, including a specimen with duplicated parameters, were manufactured. FBG sensors were used to measure changes in the temperature of the above three specimens during the microwave curing process.

### 3.1. Manufacturing of Composite Laminates with Inserted FBG Sensors

Considering the internal size of the microwave curing equipment (VHM 180/200, Vötsch Industrietechnik corp., Reiskirchen, Germany), three types of specimens (10, 15, and 20 plies) were manufactured to check the curing characteristics according to the thickness of the composite laminates, as shown in Figure 3. Four FBG (FBGKOREA corp., Daejeon, South Korea) temperature sensors were inserted at three points along the thickness direction of each specimen, as shown in Table 1. The manufactured temperature sensors used 20 mm-long glass capillary tubes, as shown in Figure 2, and all of the laminates used in this paper were based on a carbon/epoxy fabric composite material (SKYFLEX, SK Chemicals corp., Seongnam, South Korea).

The fiber optic temperature sensor (FOT) provided as part of the curing equipment was used as the reference temperature sensor to control the microwave curing cycle. The FOT sensor (TS2, Optocon corp., Dresden, Germany) is based on the absorption and transmission properties of gallium arsenide (GaAs) crystal semiconductors and has a resolution of about 0.4 nm/Kelvin. When the temperature of the semiconductor changes, an absorption shift of propagated light at the GaAs- crystal occurs. Then, the temperature change can be monitored by using an optical spectrum analyzer to measure the absorption shift [24].

To measure the temperature of the specimen surface, the FOT was attached to the surface of a vacuum bagging film made of polyimide (UPILEX-RN, UBE corp., Tokyo, Japan), which was the intermediate material for curing.

In the above table, “{}” refers to the layer in which the FBG sensor is inserted, while the subscript refers to the insertion position (C: center, S: side).

### 3.2. Cure Monitoring of Composite Laminates Using Microwaves

The autoclave method, which is the most widely used method of curing high-quality composite materials, applies the curing cycle as shown in Figure 4 to cure composite materials. The cycle consists of the following four steps: primary heating to heat the specimen from normal temperature to 80 °C (Section A); primary retention to maintain the temperature for 30 min (Section B); secondary heating to heat the specimen again to 130 °C (Section C); and secondary retention to maintain the temperature for 70 min (Section D). After completing these four steps, the specimen is cooled to complete the curing. As the two heating steps involved in the autoclave curing method use indirect heat transfer by convection, the temperature slope is set small at about ~2 °C/min to prevent overshoot by overheating in the two heating sections.

Conversely, the temperature of the specimen increases quickly during microwave curing because the process uses direct irradiation by electromagnetic waves on the dielectric carbon fiber specimen. This is the main reason why microwave curing has emerged as the next-generation curing process, as the reduced curing time can lower the curing cost, which accounts for a large portion of the cost of producing composite materials. However, because microwaves have an electromagnetic property, they are sensitive to changes in size and thickness of composite materials due to the limited depth of penetration of electromagnetic waves and the concentration of electromagnetic waves in a specific position according to the shape of the materials, and this can affect the curing quality of a material. Therefore, to ensure a specific level of quality during microwave curing of a carbon fiber reinforced composite material, it is necessary to thoroughly analyze the effects of changes in the size and thickness of a material on the curing quality.

For this study, to evaluate the effect of variations in thickness and size on the quality of microwave curing, the curing of composite specimens was monitored using FBG sensors, as shown in Figure 5. FBG temperature sensors were inserted into the specimen to measure changes in temperature at each step of the curing process. These sensors were connected to an FBG sensor interrogator (IFIS110, Fiberpro corp., Daejeon, South Korea) via a hole in the utilized equipment. Details of the interrogation system are shown in Table 2. The Bragg wavelength of each sensor was measured at 1 Hz sampling rate, and the measured signal was converted to temperature by signal processing using Equations (2) and (3). Meanwhile, temperature data from the FOT sensor were measured at 0.2 Hz sampling rate.

## 4. Test Results and Discussion

In this study, to check the effects of changes in the thickness of a composite material during microwave curing based on 75% of maximum microwave power, cure monitoring was performed using FOT and FBG sensors.

Figure 6 shows the results of temperature monitoring during the curing process of specimen 1 (thickness: 10 plies, size: 500 mm × 500 mm). This figure shows temperature measurements obtained from the reference FOT temperature sensor and the FBG temperature sensors inserted into each layer of the specimen. The FBG temperature sensors obtained temperature measurements up to about 10 °C higher than the reference temperature sensor (FOT) attached to the surface of the specimen. There were large overshoots in sections A and C, i.e., the sections in which the temperature increased, on the side but not at the center of the specimen. Therefore, the temperature did not follow the reference temperature measured by FOT, even in the temperature-retaining sections B and D.

In view of power consumption, a constant power was applied in section A and C. On the contrary, the imposed power was continuously on and off in the temperature-retaining sections B and D.

As can be seen in Figure 7, the other four specimens showed patterns similar to those presented in Figure 6. For more detailed analysis, the effects of changes in each factor on the measured temperature were quantitatively analyzed.

Since microwaves are a type of electromagnetic wave, the property and thickness of an irradiated specimen affect the penetration performance of the microwaves and thus have a direct effect on the curing quality. Therefore, it is necessary to evaluate the impact of microwaves on the curing quality of a composite material according to changes in the thickness.

Three thicknesses of 10 piles, i.e., the most widely used stack in the manufacturing of a commercial composite material structures, and 15 piles and 20 piles were specified as the parameters related to changes in the specimen thickness. Analysis of the time taken for temperature to increase in heating sections A and C showed that the time taken in section A increased by about 40% for the thickness of 20 piles, unlike the thickness of 10 piles and 15 piles; however, section C showed a specific pattern regardless of changes in thickness, as shown in Figure 8. Section A, in which the matrix was liquefied and dispersed throughout the specimen, was affected by the temperature increase when the thickness was larger (at 15 plies), while changes in thickness had no impact in section C, where the laminate was hardened, which indicates the effect of phase change of the matrix in the composite material and the associated complex correlation with electromagnetic waves.

Meanwhile, energy consumptions for all specimens are at a similar level, as shown in Figure 9. However, the energy consumption per volume is decreased, comparing with the value of 10 plies (thickness of 2.45 mm), by about 36% and 51% for the thickness of 15 (thickness of 3.67 mm) and 20 (thickness of 5.05 mm) plies, respectively. Thus, as the volume of composite gets larger, the efficiency of microwave curing is better. In this, the power supplied to the microwave device is used to analyze the energy consumption.

Changes in thickness of laminates lead to changes in heat capacity of the whole specimen, and thus can directly affect the curing quality by increasing the impact on overshoot. Evaluating the effect of changes in thickness on overshoot is very important because microwaves tend to concentrate on the edge of a structure. Quantitative analysis of overshoot in heating sections A and C, shown in Figure 10, indicates that the overshoot was larger in section C than in section A at the center of the specimen, while it was larger in section A than in section C at the side of the specimen. Overshoot caused by increase in thickness decreased slightly or remained the same at the center of the specimen, but increased at the sides in both sections A and C. Considering that the temperature of the FOT attached to the surface of the outermost layer of the specimen was the reference temperature, these results can be explained by the fact that the natural cooling effect was relatively small inside the specimen, while the heat capacity of the whole specimen increased as the thickness increased, because electromagnetic waves tend to concentrate at the edge. This result confirmed that the effects of changes in the specimen thickness on overshoot during microwave curing of the composite material were minimal at the center of the composite material specimen and very large at its sides.

The large temperature variation between the laminate layers during the curing process can inflict thermal shock on the specimen and have a negative effect on curing quality. Figure 11 shows the temperature differences in the thickness direction on the top, middle, and bottom layers of the specimen. This shows that the pattern of temperature differences in the thickness direction decreases as the specimen thickness increases in all sections, i.e., A, B, C, and D. This result is attributed to the fact that the temperature was relatively insensitive to external natural cooling because the increase in thickness meant an overall increase in the heat capacity of the whole specimen. The temperature difference was relatively larger in sections C and D than in sections A and B, which is attributed to the fact that natural cooling of the specimen increased the differences in temperature at the surface of the specimen and at the center in sections C and D, where the matrix hardened to become more solid than in sections A and B, where the temperature equilibrium of the layers occurred quickly as the matrix liquefied. As a result, the temperature gap between the layers decreased as the laminate thickness increased.

For analysis of the curing quality with changes in thickness of the laminate, an optical microscope (AM4515T8, Dino-Lite Digital Microscope corp., Hsinchu, Taiwan) of 900 × magnification was used to take cross section images at both the center (Figure 12) and side (Figure 13) of the specimen; images from two areas were compared. In the images in Figure 11, which were captured in the centers of the three specimens with different thicknesses, irregularities like voids are not observed. However, in Figure 12, voids can be easily observed in the sides of all specimens; the sizes of the voids seem to be larger as the thickness increases. That is to say, temperature overshoot caused by increases in thickness decreases the viscosity of the matrix and finally affects the curing quality of the composite, especially at the sides of the specimens.

## 5. Conclusions

To evaluate the effect of variations in thickness during the microwave curing of carbon fiber reinforced composite laminates, this study monitored the curing of composite specimens using FBG sensors. The quantitative analysis conducted by monitoring the required time and overshoot of the temperature increase at each position of the specimen when varying the specimen thickness yielded the following conclusions.

First, the time taken for the temperature to increase according to the thickness of the composite laminates increased for the thicker (15 or more piles) material in section A, but remained almost constant for section C. The time taken for the temperature to increase according to the size of the composite material laminate linearly decreased as the size increased in section A, but remained almost constant in section C.

Second, the overshoot according to changes in the thickness of the composite laminates reached a maximum of about 10 °C, and thus was not significant at the center of the specimen; however, overshoot was 3–4 times larger at the sides of the specimen than at the center.

Third, viscosity decrease induced by temperature overshoot during curing of the specimen can cause voids in the specimen after the end of the curing process; viscosity has a tendency to be larger as the specimen thickness increases, and it finally affects the curing quality of the composite.

In conclusion, overshoot, which is affected by specimen thickness, can affect the curing quality during microwave curing of a composite material. Therefore, to improve the quality of microwave curing of composite materials, it is necessary to optimize the curing cycle by considering the characteristics of the microwave curing equipment. Then, the characteristics of the microwave curing with various parameters have to be investigated. Furthermore, the mechanical properties of the composites cured by microwave devices will be evaluated after optimization of the curing cycle by simulating microwave curing.

## Figures and Tables

**Figure 1 materials-13-01720-f001:**
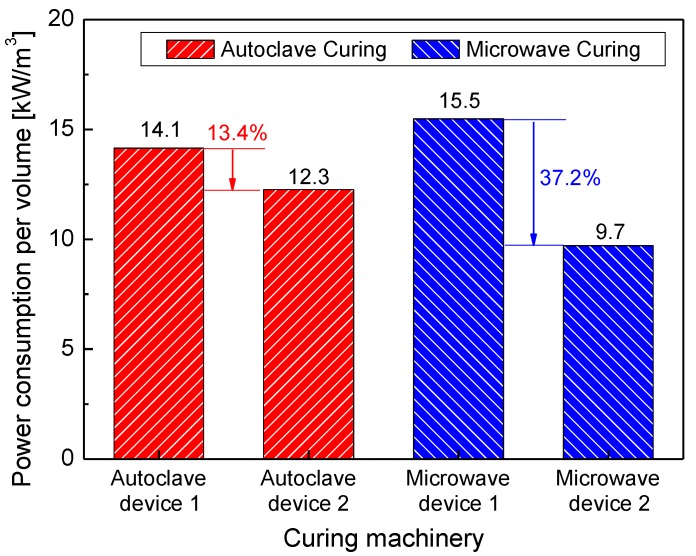
Comparison of power consumption in autoclave and microwave curing machinery.

**Figure 2 materials-13-01720-f002:**
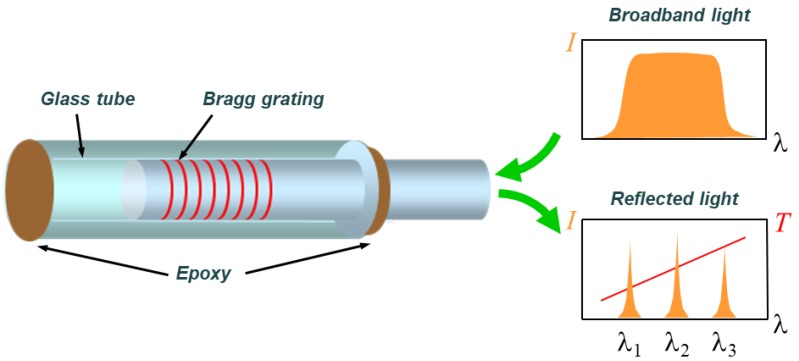
Configuration of a fiber Bragg grating (FBG) temperature sensor [19].

**Figure 3 materials-13-01720-f003:**
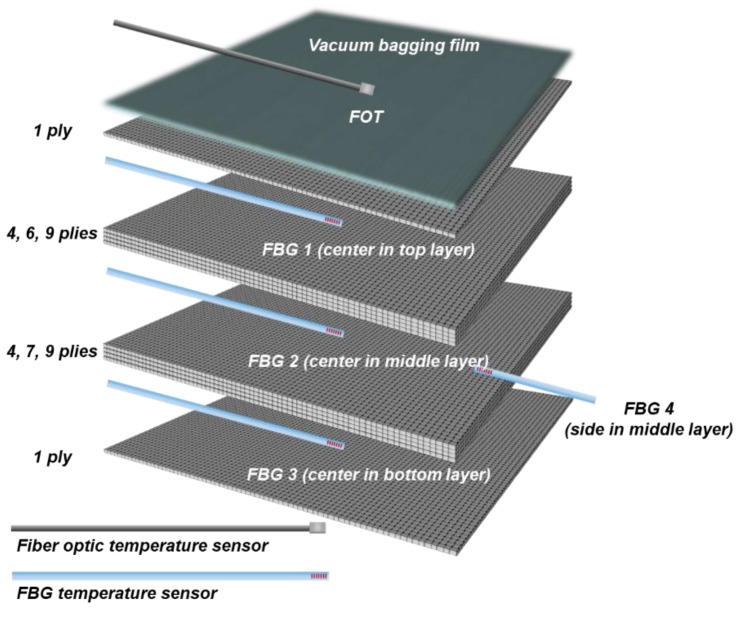
Configuration of composite specimens with sensor locations.

**Figure 4 materials-13-01720-f004:**
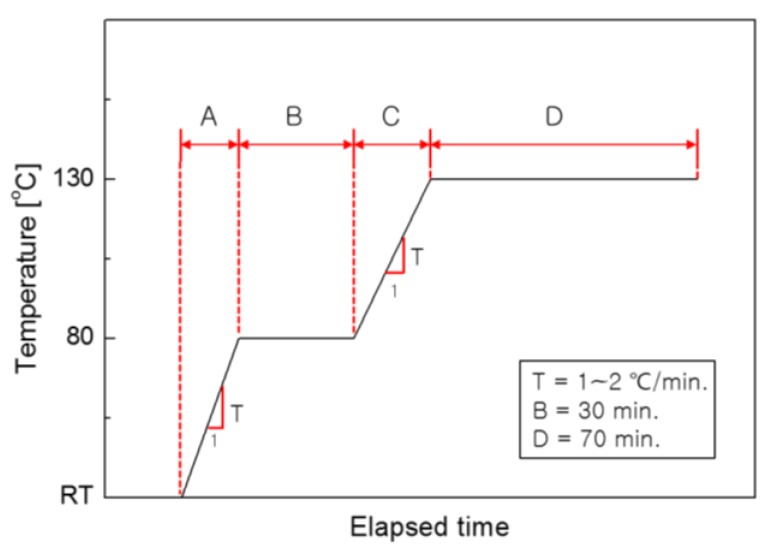
Typical curing cycle of the autoclave method.

**Figure 5 materials-13-01720-f005:**
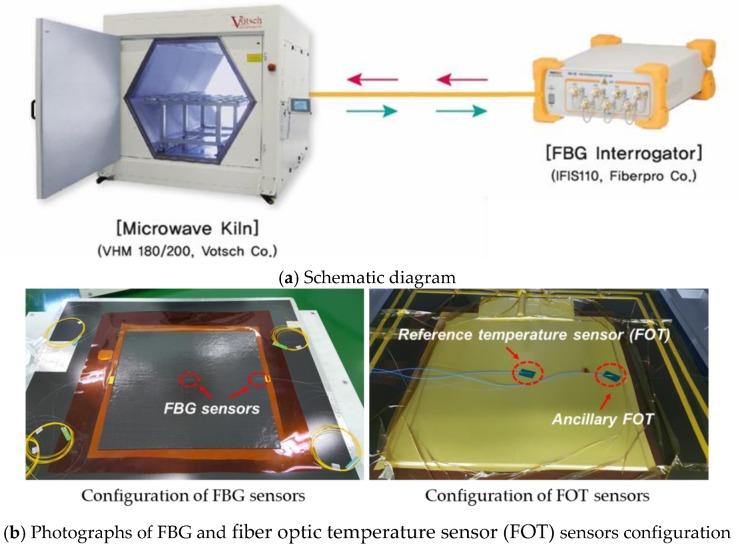
Schematic diagram of the experimental setup including photos of the specimen.

**Figure 6 materials-13-01720-f006:**
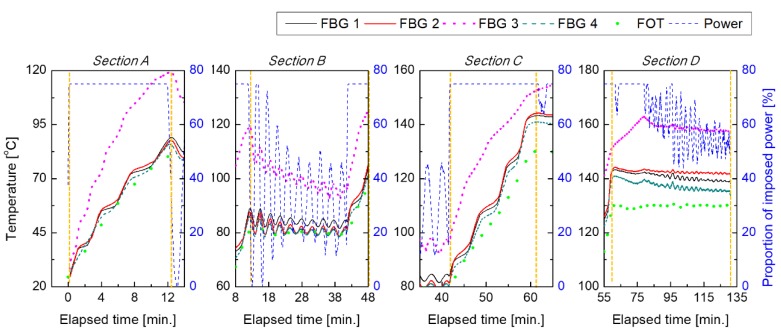
Temperature readings obtained from all sensors during cure monitoring of specimen 1 in 1 S/s and 0.2 S/s sampling rates of FBG sensors and FOT sensor, respectively; (Section A) primary heating, (Section B) primary retention, (Section C) secondary heating, (Section D) secondary retention.

**Figure 7 materials-13-01720-f007:**
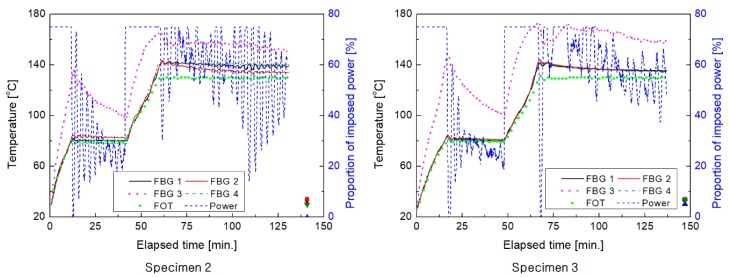
Results of cure monitoring of the other specimens (2 (left), 3(right)).

**Figure 8 materials-13-01720-f008:**
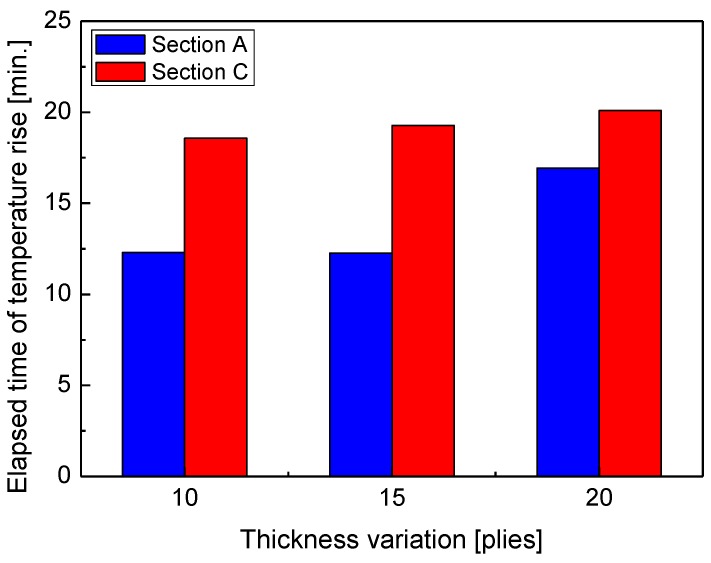
Elapsed time of temperature rise with thickness variation.

**Figure 9 materials-13-01720-f009:**
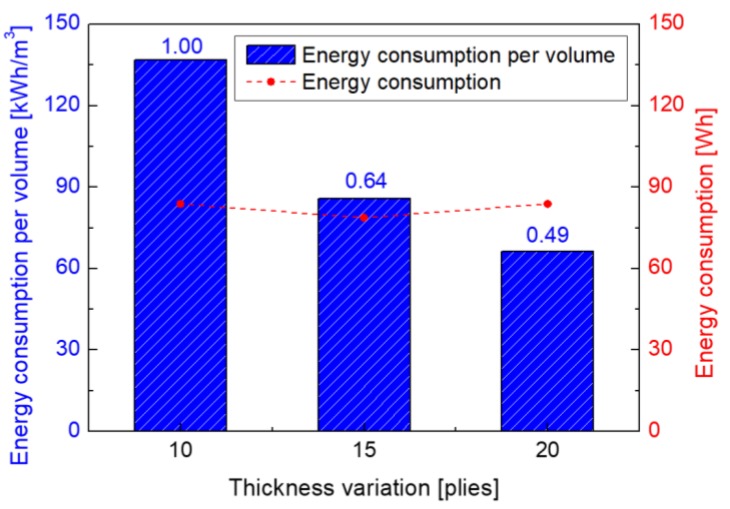
Energy consumption per volume with thickness change of composite.

**Figure 10 materials-13-01720-f010:**
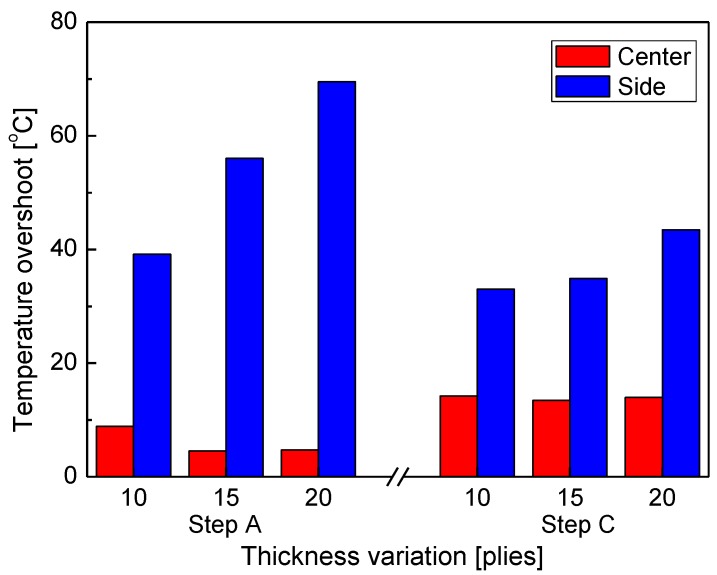
Temperature overshoot at the center and sides of the specimens with thickness variation.

**Figure 11 materials-13-01720-f011:**
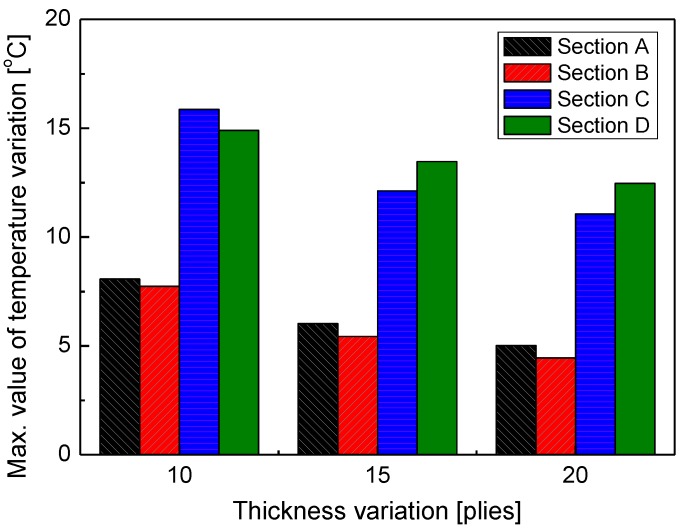
Maximum temperature difference through thickness with thickness variations.

**Figure 12 materials-13-01720-f012:**
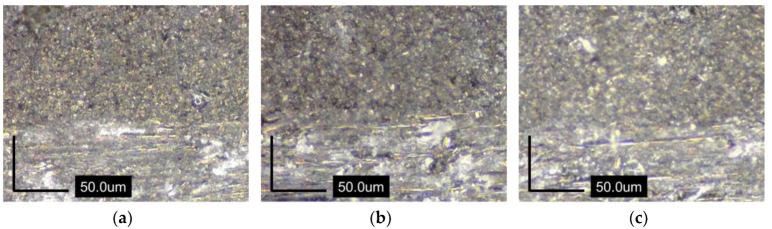
Cross section images in the center of specimens: (**a**) 10 plies; (**b**) 15 plies; (**c**) 20 plies.

**Figure 13 materials-13-01720-f013:**
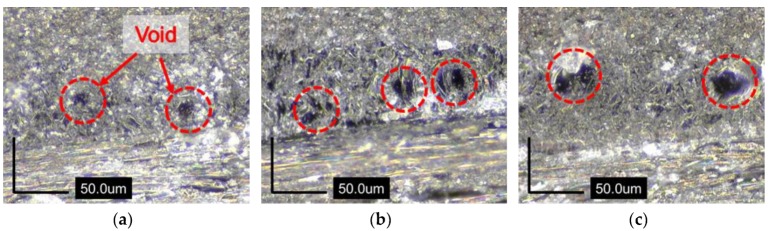
Cross section images in the side of specimens: (**a**) 10 plies; (**b**) 15 plies; (**c**) 20 plies.

**Table 1 materials-13-01720-t001:** Information on the composite laminates and the insertion of fiber Bragg grating (FBG) sensors.

Thickness Variation	Area (m^2^)	Layers	Stacking Sequence with Embedded Sensors
Specimen 1	500 × 500	10 plies	[Fabric/{0}_C_/Fabric_4_/{0}_C,S_/Fabric_4_/{0}_C_/Fabric]_T_
Specimen 2		15 plies	[Fabric/{0}_C_/Fabric_6_/{0}_C,S_/Fabric_7_/{0}_C_/Fabric]_T_
Specimen 3		20 plies	[Fabric/{0}_C_/Fabric_9_/{0}_C,S_/Fabric_9_/{0}_C_/Fabric]_T_

**Table 2 materials-13-01720-t002:** Specifications of an FBG interrogation system.

IFIS-110 FBG Interrogator	
Wavelength range	85 nm (1510–1595 nm)
Wavelength accuracy	<20 pm (0–50 °C)
Wavelength repeatability	<3 pm
Resolution	1 pm
Max. sampling rate	200 Hz (for 1 CH)
Max. # of channels	7

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
