# Peer review of "Microwave Curing Characteristics of CFRP Composite Depending on Thickness Variation Using FBG Temperature Sensors"

_materials, 2020, doi:10.3390/ma13071720_

Round 1

Reviewer 1 Report

Paper investigates the microwave curing process of CFRP composites, specifically the affect of composite thickness on the quality of the final material by monitoring the thermal variations using FBGs.

Overall the research is carried out in an acceptable manner. There is a clear problem study - degradation in quality of microwave cured composites - and the authors investigations uncover that the thermal variations are more excessive around the edges of the composites by using thermal sensing. They also find that the thickness of the composite does indeed cause an increase in the temperature overshoot and therefore decreases the quality of the final material in the form of voids.

With regards to the novelty of the work. FBG temperature sensing is well established, as is microwave curing of composites. However, I perceive uniqueness in the demonstration of relating the thickness of composites to the overshoot of temperatures and; therefore, the quality of the material. I believe this to be of acceptable originality.

The paper requires significant review/editing of writing. I found some sections to be very difficult to understand and sometimes found misinformation simply due to the writing. Some examples will be listed below.

I also believe that some extra information and work is required in some sections, as will be listed below. Following these changes, I believe the paper is worthy of acceptance.

Abstract

The abstract requires significant editing, it is a poor overview of the work. I would suggest completely changing the first sentence as it makes no sense. I also suggest adding in some more specific results - for example, temperature overshoots for each thickness/section.

Introduction

Again, improvement to the writing is required. Examples of poorly constructed sentences: Lines 47-49, 76-80. I think the whole section requires review and editing.

Figure 1: Difference between device 1 and 2 needs to be clearly identified in the figure. Difficult to ascertain the benefits of microwave curing from this figure.

I believe some background information regarding the microwave curing technique used is required. How does it work? What other techniques can be used? etc.

Section 2

Remove reference from section title.

Use a better reference for FBG principles, such as Rao - "in-fibre Bragg gratings".

Operating principle could be explained in more detail. What is the Bragg wavelength? Perhaps add to figure 2, showing that a broadband light is projected through the FBG and the reflection appears as a "peak", that shifts based on the measurands (temperature, strain etc.).

Information is required either here, or in future section, regarding the accuracy/of the FBG sensors used in this work: What is the interrogator resolution, range, repeatability etc.

How do you monitor multiple FBGs simultaneously if they are not multiplexed?

Section 3

Table 1 - not sure what the convention is, but the description of the stacking sequence is very confusing to me. Could this be made simpler?

I think more information is required regarding the material used.

Figure 5 - not clear, I think each photo should be labelled and then described separately in the caption. 

Section 4

Figure 6- Resolutions of each sensor should be identified.

Figure 6 - A, B, C and D should be described for convenience.

I think the sensors in Figure 6 should be more clearly labelled and also identified in Figure 3.

Questions

The authors are asked to comment why only the side of the middle layer was monitored, why not monitor the side of each layer? Does this large variance occur in each layer? It seems monitoring the centre of each layer was rather pointless as it doesn't change. I think it would be interesting to monitor various points around the composite, to build a sort of "map" of temperature gradients throughout the sample.

The authors are asked to compare the resolution of the FBG sensors AND the FOT sensor to the difference in temperature measurement. These values should be inserted into the paper and discussed.

The authors are asked to comment on the large variation in temperature in the centre during Step D as apposed to Step B where there is little difference.

The authors are asked to more thoroughly discuss the solution to the problem. I think this is possibly the main issue with this piece of work. I agree there is novelty in determining that the thickness of the composite affects the quality due to thermal variations; however, I feel the authors have failed to discuss or outline how this would be solved. They simply suggest "optimize curing cycle". I think the paper requires a "future work" type section before the conclusion that outlines what the outcomes of this research mean for the broader field and how the authors suggest this could be solved.

Author Response

Abstract

The abstract requires significant editing, it is a poor overview of the work. I would suggest completely changing the first sentence as it makes no sense. I also suggest adding in some more specific results - for example, temperature overshoots for each thickness/section.

Response: As the reviewer mentioned, abstract is corrected by adding specific results like overshoots and voids.

Introduction

Again, improvement to the writing is required. Examples of poorly constructed sentences: Lines 47-49, 76-80. I think the whole section requires review and editing.

Response: This paper has been proofread by a professional English calibration company.

Figure 1: Difference between device 1 and 2 needs to be clearly identified in the figure. Difficult to ascertain the benefits of microwave curing from this figure.

Response: Device 1 and 2, which are autoclave and microwave devices, have difference of effective working volume of specimen. The effective volume of device 2 is larger than that of device 1. So, it shows that reduction of power consumption per volume of microwave device is higher than that of autoclave.

I believe some background information regarding the microwave curing technique used is required. How does it work? What other techniques can be used? etc. 

Response: As the reviewer mentioned, explanation of microwave curing technique is added in the section of introduction.

Section 2

Remove reference from section title.

Use a better reference for FBG principles, such as Rao - "in-fibre Bragg gratings".

Operating principle could be explained in more detail. What is the Bragg wavelength? Perhaps add to figure 2, showing that a broadband light is projected through the FBG and the reflection appears as a "peak", that shifts based on the measurands (temperature, strain etc.).

Information is required either here, or in future section, regarding the accuracy/of the FBG sensors used in this work: What is the interrogator resolution, range, repeatability etc.

How do you monitor multiple FBGs simultaneously if they are not multiplexed?

Response: As the reviewer mentioned, reference in section title is removed. And for improving readability, section 2 and 3.2 is corrected overall.

Section 3

Table 1 - not sure what the convention is, but the description of the stacking sequence is very confusing to me. Could this be made simpler?

I think more information is required regarding the material used.

Figure 5 - not clear, I think each photo should be labelled and then described separately in the caption.

Response: As the reviewer mentioned, Table 1 and Figure 5 are corrected for simplicity of expression.

Section 4

Figure 6- Resolutions of each sensor should be identified.

Figure 6 - A, B, C and D should be described for convenience.

I think the sensors in Figure 6 should be more clearly labelled and also identified in Figure 3.

Response: Resolutions of sensors depend on data logger. Therefore, sampling rate of each sensor is inserted to caption in both Figure 6 and section 3.2. Also, the entire sections during the cure process are described in the Figure. Furthermore, Figure 3 and 6 are corrected for enhancing readability of labels.

Questions

The authors are asked to comment why only the side of the middle layer was monitored, why not monitor the side of each layer? Does this large variance occur in each layer? It seems monitoring the centre of each layer was rather pointless as it doesn't change. I think it would be interesting to monitor various points around the composite, to build a sort of "map" of temperature gradients throughout the sample.

Response: Characteristics of microwave curing with thickness variation of CFRP were evaluated in this study. Microwave curing, which is method of directly heating overall specimen, can cause high degradation of composite quality on side (edge) region. Therefore, side and center region of specimen were evaluated and analyzed using cure monitoring in real time. If cure monitoring is conducted on multi-layers of composite, many sensors have to be inserted in laminate. Because that can cause structure change of laminate, 4 FBG sensors were applied to the composite.

The authors are asked to compare the resolution of the FBG sensors AND the FOT sensor to the difference in temperature measurement. These values should be inserted into the paper and discussed.

Response: According to the previous response, sampling rates of FBG and FOT sensors are inserted for better understanding.

The authors are asked to comment on the large variation in temperature in the centre during Step D as apposed to Step B where there is little difference.

The authors are asked to more thoroughly discuss the solution to the problem. I think this is possibly the main issue with this piece of work. I agree there is novelty in determining that the thickness of the composite affects the quality due to thermal variations; however, I feel the authors have failed to discuss or outline how this would be solved. They simply suggest "optimize curing cycle". I think the paper requires a "future work" type section before the conclusion that outlines what the outcomes of this research mean for the broader field and how the authors suggest this could be solved.

Response: As the reviewer mentioned, curing cycle of autoclave curing was applied to microwave curing in this study. Therefore, the curing cycle can fundamentally be reason for occurring overshoot and voids. To overcome that, future work was commented in conclusion section.

Reviewer 2 Report

Review Materials 725234
Microwave Curing Characteristics of CFRP composite depending
on thickness variation using FBG Temperature Sensors

General content :

Through various measurements and devices, the authors have highlighted the problem of process optimization. The chemistry and physics can explain and quantify the amount of energy needed for the curing of a given composite material. The authors must argue in that way in the beginning of the paper and clarify their "power point of view". The processing of the curing itself induces some rapid variations of viscosity and degassing which can cause voids in the final composite material. This was clearly highlighted in Figure 11 and 12. This paper shows the challenge of temperature mastering using microwave, and it is clearly summarized in the conclusion.

General comments :

  • In the "energy point of view", the efficiency can be quantified by the ratio between the needed energy for the curing (kW.h output) to the electrical energy (kW.h input) used as input of the heating device (autoclave or microwave for instance). These energies can advantageously be normalized relatively to the volume resulting in an energy per volume (kW.h/m3). This can be discussed by the authors in terms of efficiency of the energy conversion from electricity to heat.
  • In the "power point of view", which seems to be that of the authors is based on "time is money", so that (lines 155 to 157) "This is the main reason why microwave curing has emerged as the next-generation curing process, as the reduced curing time can lower the curing cost, which accounts for a large portion of the cost of producing composite materials.". In that point of view, since the energy needed for the curing is known (kW.h/m3), the power (kW) dedicated to curing of the material is to be maximized is order to reduce the processing time. Nevertheless, this is a restricted view of the chemistry and physics of the curing process (see [Wei, 1993]). Thus, an analysis of the effective physical properties is carried out by the authors, and an optimization is to be lead in the end.

Some additional references must be provided such as :

[Wei, 1993] Jianghua Wei, Martin C. Hawley, John D. Delong, Mark Demeuse, "Comparison of microwave and thermal cure of epoxy resins", Polymer Engineering & Science, Vol. 33, N. 17, p. 1132-1140, 1993.

[Baskaran, 2014] M. Baskaran, I. Ortiz de Mendibil, M. Sarrionandia, J. Aurrekoetxea, J. Acosta, U. Argarate, D. Chico, "Manufacturing cost comparison of RTMHP, RTM and CRTM for an automotive roof" ECCM16 - 16th European Conference on Composite Materials, Seville, Spain, 22-26 June 2014.

[Teufl, 2018] Daniel Teufl and Swen Zaremba, "2.45 GHz Microwave Processing and Its Influence on Glass Fiber Reinforced Plastics", Matrials, 2018, 11, 838.

Questions and requested corrections :
* Lines 51 to 53 :
"In previous studies on the microwave curing of composite materials, Yarlagadda and Hsu [6] reported that 2.45GHz was the most effective frequency band for the curing of composite materials, and that frequency band has been mostly used in microwave curing since then."
The authors must reformulate their sentence since the cited work only says "The magnetron microwave generator used in this research is operating at a frequency of 2.45 GHz.". It do not mean that this frequency is "the most effective frequency band for the curing of composite materials", but simply that this frequency band was available of commercialized microwave ovens.

  • Lines 66 to 75 :
    These comments are to be reformulated making it clear the difference between power and energy needed for curing a unit volume of 1 m3. More explicitly, is the mentioned power an average value or a maximum value ? Generally, the needed power is increasing with the temperature (see [Teufl, 2018], Figure 5). This is well explained for the autoclave method (lines 144 to 152) and the microwave curing (lines 153 to 163).
  • * Line 96 :
    "Figure 1. Comparison of power consumptions in autoclave and microwave curing machinery "
    The power consumption per volume (W/m3) is an estimator of the needed power, but it is a partial view of the energy provision of the curing machinery device. In an electrical power balance, the maximum power (W) is necessary to dimension the power cable area, but the cost in terms of energy is the product of the power by the time of use, and expressed in J (=W.s) or more commonly in W.h (=3600J). The power is one thing, the energy is an other resulting from the product with time and this is the real consumption cost for the manufacturer. The authors must argue for the costs and cite a work on this point of view (see [Baskaran, 2014], Table 5).

Author Response

Questions and requested corrections :

* Lines 51 to 53 :

"In previous studies on the microwave curing of composite materials, Yarlagadda and Hsu [6] reported that 2.45GHz was the most effective frequency band for the curing of composite materials, and that frequency band has been mostly used in microwave curing since then."

The authors must reformulate their sentence since the cited work only says "The magnetron microwave generator used in this research is operating at a frequency of 2.45 GHz.". It do not mean that this frequency is "the most effective frequency band for the curing of composite materials", but simply that this frequency band was available of commercialized microwave ovens.

Response: As the reviewer mentioned, frequency of microwave ovens is 2.45GHz in this paper. So, the paragraph is corrected with various references.

* Lines 66 to 75 :

These comments are to be reformulated making it clear the difference between power and energy needed for curing a unit volume of 1 m3. More explicitly, is the mentioned power an average value or a maximum value ? Generally, the needed power is increasing with the temperature (see [Teufl, 2018], Figure 5). This is well explained for the autoclave method (lines 144 to 152) and the microwave curing (lines 153 to 163).

Response: In order to compare power consumption between autoclave and microwave facility, power consumption per effective working volume of all devices was represented by heating power. Of these, circulating fan motor power consumption of autoclave is added. The paragraph was not only corrected for better understanding, the reference [Teufl, 2018] was also added.

* Line 96 :

"Figure 1. Comparison of power consumptions in autoclave and microwave curing machinery "

The power consumption per volume (W/m3) is an estimator of the needed power, but it is a partial view of the energy provision of the curing machinery device. In an electrical power balance, the maximum power (W) is necessary to dimension the power cable area, but the cost in terms of energy is the product of the power by the time of use, and expressed in J (=W.s) or more commonly in W.h (=3600J). The power is one thing, the energy is an other resulting from the product with time and this is the real consumption cost for the manufacturer. The authors must argue for the costs and cite a work on this point of view (see [Baskaran, 2014], Table 5).

Response: As the previous response, electrical power of the device is consumed in order to heat composite materials. Thus, the energy and cost of devices, which are related to curing time, will naturally cut if a curing cycle is decreased. The curing cycle of microwave curing has to be optimized for better curing.

Reviewer 3 Report

The paper focuses on the use of FBG technology to monitor the curing process of crfp.

The structure of the manuscript is fine and the English grammar is satisfactory.

The methodology is well described and sound. One minor issue is about the number of tests: each experiment is unique and, therefore, it is not clear how much the behavior varies, with respect to the installation and samples.

If it is not possible to perform more tests, I would encourage the Authors to mention this aspect.

In my opinion, the paper can be accepted upon minor revisions.

Author Response

The methodology is well described and sound. One minor issue is about the number of tests: each experiment is unique and, therefore, it is not clear how much the behavior varies, with respect to the installation and samples.

If it is not possible to perform more tests, I would encourage the Authors to mention this aspect.

Response: Experiment is conducted to compare temperature behaviors between center and side region because there are many behavior variations in side region of composite specimen in curing. Furthermore, to evaluation of effect on thickness of overall specimen, temperature characteristic in representative middle layer is investigated. For the future, research about optimization of curing cycle is conducted. And then it is expected that characteristics on various parameters are investigated in curing of microwave. So, paragraph about optimization of curing cycle is inserted in conclusion.

Reviewer 4 Report

1. The manuscript investigated the effect of CFRP composite thickness on the quality of microwave curing.

2. The large temperature overshoot and voids were observed in the side of specimens. How to improve it ?

3. The experimental results showed that the quality of microwave curing is significantly affected by the CFRP composite thickness in terms of temperature overshooting and temperature variation across the thickness. It may be more useful and practical that the authors can characterize the quality of microwave curing in terms of mechanical properties of CFRP composite such as Young’s modulus, tensile strength etc.

4. In the conclusion, the author stated “it is necessary to optimize the curing cycle to improve the quality of microwave curing of composite materials”. How to optimize the curing cycle?

Author Response

2. The large temperature overshoot and voids were observed in the side of specimens. How to improve it ?

Response: If optimization of curing cycle for microwave curing facility is conducted, it is expected that problems such as overshoot and voids can be overcome.

3. The experimental results showed that the quality of microwave curing is significantly affected by the CFRP composite thickness in terms of temperature overshooting and temperature variation across the thickness. It may be more useful and practical that the authors can characterize the quality of microwave curing in terms of mechanical properties of CFRP composite such as Young’s modulus, tensile strength etc.

Response: In this paper, in order to investigate characteristics of microwave curing, temperature overshoot and voids in cross section of specimen were measured. Of course, it can be another way of investigating characteristics to measure mechanical property changes as you mentioned. However, the authors focus to find such characteristics using cure monitoring methodology and mechanical property changes will be compared after curing process of microwave facility is optimized, which can show the exact mechanical property, in the future work.

4. In the conclusion, the author stated “it is necessary to optimize the curing cycle to improve the quality of microwave curing of composite materials”. How to optimize the curing cycle? 

Response: For preventing occurrence of the temperature overshoot and voids, optimization of curing cycle is key issue. For optimizing curing cycle, many parameters on microwave facility such as magnetron numbers and positions, power intensity, exposure time, etc are related. By experiments on those parameter changes or simulation using numerical analysis the curing cycle of microwave facility can be optimized.

Reviewer 5 Report

In this paper, the authors reported using FBG temperature sensors for evaluating the effects of variation in thickness on the quality of microwave curing,  the experiment confirmed that microwave curing reduced the curing time in the sections where the temperature rose and that thickness of the specimen affected the overshoot, the article is very well written, clear, concise, and suitable for the scope of the journal. Only several suggestions:

  1. In my mind, FBG technology obtains the wavelength shift, then through calibration obtain the change in temperature, please give more detail about how u obtain the temperature value.
  2. Suggest improving the font in Fig.11 and Fig.12.
  3. Please give more detail about the fiber optic temperature sensor (FOT) used for reference.
  4. Suggestion reference to one typical literature for the FBG theory part. Such as:

 Erdogan, T. (1997). Fiber grating spectra. Journal of lightwave technology, 15(8), 1277-1294.

Hill, K. O., & Meltz, G. (1997). Fiber Bragg grating technology fundamentals and overview. Journal of lightwave technology, 15(8), 1263-1276.

Kashyap, R. (2009). Fiber Bragg gratings. Academic Press.

Othonos, A. (1997). Fiber Bragg gratings. Review of scientific instruments, 68(12), 4309-4341.

Author Response

1. In my mind, FBG technology obtains the wavelength shift, then through calibration obtain the change in temperature, please give more detail about how u obtain the temperature value.

Response: A general temperature function of FBG modulation was used to measure the temperature. In order to obtain exact temperature in the specimen, we used 2 optical sensors. One is FOT, which is originally equipped in the microwave facility, and it indicates exact temperature. And, FBG sensors can measure exact in-situ temperature by measuring temperature changes and calibrating it using those from FOT, as you mentioned,

2. Suggest improving the font in Fig.11 and Fig.12.

Response: As the reviewer mentioned, the font is corrected for better expression

3. Please give more detail about the fiber optic temperature sensor (FOT) used for reference.

Response: As the reviewer mentioned, details on FOT are inserted in section 3.1.

4. Suggestion reference to one typical literature for the FBG theory part. Such as:

Response: As the reviewer mentioned, report of the FBG theory is modified using just one reference.

Round 2

Reviewer 2 Report

Questions and requested corrections:

* Lines 51 to 53:

"In previous studies on the microwave curing of composite materials, Yarlagadda and Hsu [6] reported that 2.45GHz was the most effective frequency band for the curing of composite materials, and that frequency band has been mostly used in microwave curing since then."

The authors must reformulate their sentence since the cited work only says "The magnetron microwave generator used in this research is operating at a frequency of 2.45 GHz.". It does not mean that this frequency is "the most effective frequency band for the curing of composite materials", but simply that this frequency band was available of commercialized microwave ovens.

Authors’ response: As the reviewer mentioned, frequency of microwave ovens is 2.45GHz in this paper. So, the paragraph is corrected with various references.

Reviewer’s response: This 2.45 GHz frequency is not "the most effective frequency band for the curing of composite materials". The authors of the work they cite are not saying that ! This frequency range is used since it was commercially available at the early beginning of this microwave technology.

* Lines 66 to 75:

These comments are to be reformulated making it clear the difference between power and energy needed for curing a unit volume of 1 m3. More explicitly, is the mentioned power an average value or a maximum value? Generally, the needed power is increasing with the temperature (see [Teufl, 2018], Figure 5). This is well explained for the autoclave method (lines 144 to 152) and the microwave curing (lines 153 to 163).

Authors’ response: In order to compare power consumption between autoclave and microwave facility, power consumption per effective working volume of all devices was represented by heating power. Of these, circulating fan motor power consumption of autoclave is added. The paragraph was not only corrected for better understanding, the reference [Teufl, 2018] was also added.

Reviewer’s response: As suggested, the reference [Teufl, 2018] was added, but the comments were not enriched as much as required in view to clarify the power needs (kW/m3) and the energy consumption (kW.h/m3) of these fabrication process.

* Line 96:

"Figure 1. Comparison of power consumptions in autoclave and microwave curing machinery "

The power consumption per volume (W/m3) is an estimator of the needed power, but it is a partial view of the energy provision of the curing machinery device. In an electrical power balance, the maximum power (W) is necessary to dimension the power cable area, but the cost in terms of energy is the product of the power by the time of use, and expressed in J (=W.s) or more commonly in W.h (=3600J). The power is one thing, the energy is an other resulting from the product with time and this is the real consumption cost for the manufacturer. The authors must argue for the costs and cite a work on this point of view (see [Baskaran, 2014], Table 5).

Response: As the previous response, electrical power of the device is consumed in order to heat composite materials. Thus, the energy and cost of devices, which are related to curing time, will naturally cut if a curing cycle is decreased. The curing cycle of microwave curing has to be optimized for better curing.

Reviewer’s response: The authors argue for the curing time which may decrease, but they do not quantify it in the present case. That is the interest of such a study, i.e. being clear both on the needs of these fabrication process:

* the needed power (kW/m3)

* the energy consumption (kW.h/m3).

Author Response

Questions and requested corrections :

* Lines 51 to 53 :

"In previous studies on the microwave curing of composite materials, Yarlagadda and Hsu [6] reported that 2.45GHz was the most effective frequency band for the curing of composite materials, and that frequency band has been mostly used in microwave curing since then."

The authors must reformulate their sentence since the cited work only says "The magnetron microwave generator used in this research is operating at a frequency of 2.45 GHz.". It do not mean that this frequency is "the most effective frequency band for the curing of composite materials", but simply that this frequency band was available of commercialized microwave ovens.

Response: As the reviewer mentioned, frequency of microwave ovens is 2.45GHz in this paper. So, the paragraph is corrected with various references.

Reviewer’s response: This 2.45 GHz frequency is not "the most effective frequency band for the curing of composite materials". The authors of the work they cite are not saying that ! This frequency range is used since it was commercially available at the early beginning of this microwave technology.

Response: As the reviewer mentioned, for improving readability and understanding of meaning, the previous expression is corrected to “2.45GHz is the most popular frequency in commercial device for the curing of composite materials”.

* Lines 66 to 75 :

These comments are to be reformulated making it clear the difference between power and energy needed for curing a unit volume of 1 m3. More explicitly, is the mentioned power an average value or a maximum value ? Generally, the needed power is increasing with the temperature (see [Teufl, 2018], Figure 5). This is well explained for the autoclave method (lines 144 to 152) and the microwave curing (lines 153 to 163).

Response: In order to compare power consumption between autoclave and microwave facility, power consumption per effective working volume of all devices was represented by heating power. Of these, circulating fan motor power consumption of autoclave is added. The paragraph was not only corrected for better understanding, the reference [Teufl, 2018] was also added.

Reviewer’s response: As suggested, the reference [Teufl, 2018] was added, but the comments were not enriched as much as required in view to clarify the power needs (kW/m3) and the energy consumption (kW.h/m3) of these fabrication process.

Response: In analyzing power consumption of autoclave and microwave curing, the same curing cycle is applied and constant powers are supplied to exclude fluctuation of power during composite curing occurred in the reference. Under these conditions, power consumptions of devices are discussed in the section of introduction. Moreover, the results from this study of imposed power proportion and energy consumption per volume are expressed in Figs. 6, 7 and Fig. 9, respectively. In this, power of the microwave device is used to analyze the energy consumption.

* Line 96 :

"Figure 1. Comparison of power consumptions in autoclave and microwave curing machinery "

The power consumption per volume (W/m3) is an estimator of the needed power, but it is a partial view of the energy provision of the curing machinery device. In an electrical power balance, the maximum power (W) is necessary to dimension the power cable area, but the cost in terms of energy is the product of the power by the time of use, and expressed in J (=W.s) or more commonly in W.h (=3600J). The power is one thing, the energy is an other resulting from the product with time and this is the real consumption cost for the manufacturer. The authors must argue for the costs and cite a work on this point of view (see [Baskaran, 2014], Table 5).

Response: As the previous response, electrical power of the device is consumed in order to heat composite materials. Thus, the energy and cost of devices, which are related to curing time, will naturally cut if a curing cycle is decreased. The curing cycle of microwave curing has to be optimized for better curing.

Reviewer’s response: The authors argue for the curing time which may decrease, but they do not quantify it in the present case. That is the interest of such a study, i.e. being clear both on the needs of these fabrication process:

 * the needed power (kW/m3)

 * the energy consumption (kW.h/m3).

Response: As the reviewer mentioned, power consumptions are expressed in Figures 6 and 7. 75% of the maximum power (fixed value) has been continuously on and off in overall microwave curing cycle. And, power consumption fluctuating during microwave curing is used to calculate the energy consumption of the microwave device. From the results, the energy consumptions (W.h) for all specimens show similar level. However, the energy consumption per volume (kW.h/m3) is decreased as the thickness of specimen increases. Thus, as the volume of composite gets larger, the efficiency of microwave curing is better. These discussions are expressed in section 4 (Line 240~245) and Figure 9. This figure is inserted for the discussion about energy consumption of microwave curing.

Reviewer 4 Report

The manuscript can be published as it is.

Author Response

Thanks a lot for your review.

Reviewer 5 Report

The authors reply to my comments well, so now I can recommend it publish.

Author Response

Thanks a lot for your review.

Round 3

Reviewer 2 Report

  • Most of the remarks have been taken into account.
  • The requested additional references in the first review will be of great interest for the reader.